# Associations between objective and self-perceived physical activity and participation in everyday activities in mild stroke survivors

**Cristina de Diego-Alonso**[1]☉*, **Jorge Alegre-Ayala**[2]‡, **Julia Blasco-Abadía**[1]‡,
**Víctor Doménech-García**[1]‡, **Part&Sed-Stroke collaborators'**[¶], **Pablo Bellosta-López**[1]☉

**1** Universidad San Jorge, Campus Universitario, Villanueva de Gállego, Zaragoza, Spain, **2** Centro de Neurorrehabilitación intensiva CIRONLAB, Valladolid, Castilla y León, Spain

¶ Part&Sed-Stroke collaborators' is provided in the Acknowledgments.
☉ These authors contributed equally to this work.
‡ JA-A, JB-A and VD-G also contributed equally to this work.
* cdediego@usj.es

## Abstract

### Background and purpose

Stroke survivors present limited levels of physical activity (PA) and participation in everyday activities although the specific interaction between PA and participation in these individuals is still uncertain. This study aimed to analyse the relationship between PA and participation in everyday activities among Spanish mild stroke survivors.

### Methods

A total of 130 mild stroke survivors (61.3 ± 12.4 years, 35% female) with preserved walking ability and without cognitive and communication impairments participated in this cross-sectional study involving several rehabilitation centres from Spain. Self-reported levels of PA were reported by the International Physical Activity Questionnaire - short form (IPAQ-SF). Objective PA measures were monitored with the wristband Fitbit Inspire 2, recording the average steps/day and kilocalories/day. Participation and activity satisfaction levels were measured with the Satisfaction with Daily Occupations-Occupational Balance (SDO-OB) and participation retention through Activity Card Sort (ACS).

### Results

ACS total score showed a weak correlation with self-reported PA (rho = 0.324) and moderate correlations with kilocalories/day and average steps/day (rho ≥ 0.581), while stronger correlations were found for the ACS subdomain of instrumental activities (rho ≥ 0.640) compared to the subdomains of leisure activities and social participation (rho ≤ 0.454). SDO-OB participation showed moderate correlations with kilocalories/day, and average steps/day (rho ≥ 0.647), and a weak correlation with self-reported PA (rho = 0.303). Weaker correlations were found for SDO-OB satisfaction with objective PA measures (rho = 0.407)

**Data availability statement:** All relevant data are within the manuscript and its Supporting Information files.

**Funding:** This study was supported by a research award grant from the "Colegio Oficial de Terapeutas Ocupacionales de la Comunidad de Madrid" (COPTOCAM). Authors CDA, JBA, VDG and PBL are members of the research group MOTUS supported by "Gobierno de Aragón" (n. B60_23D). JBA is supported by the Grant PIF 2022-2026 from "Gobierno de Aragón", Spain. The funders did not have any role in this study.

**Competing interests:** The authors have declared that no competing interests exist.

and self-reported PA (rho = 0.254). Relationships between variables were explored by calculating Spearman correlation coefficients.

## Discussion and conclusions

The objective and self-reported measures of PA in mild stroke survivors have a bilateral relationship with their current participation levels and the retained instrumental activities of daily living. However, the weaker correlations with leisure and social participation may suggest that promoting PA alone without integrating it into daily activities relevant to the stroke survivor may be insufficient to achieve comprehensive goals during rehabilitation programs.

## Introduction

Stroke represents the second highest risk of death and the third source of disability worldwide [1]. An estimated 1.5 million people in Europe will suffer a stroke in 2025, with a 27% increase in survival rate [2]. Consequently, the total number of stroke survivors with sequelae in Europe will increase over the next 30 years [3]. This situation will increase economic and healthcare cost [1] unless the level of dependency is reduced by identifying modifiable multifactorial relationships linked to healthy lifestyles [4], such as regular physical activity (PA) and active participation in everyday activities.

Regular PA is a priority for stroke survivors in accordance with the World Health Organisation (WHO) [5]. However, about 70% of stroke survivors are sedentary [6], and only 17% follow the recommended PA guidelines [7], with these levels decreasing further over the subsequent years [8]. PA positively affects most modifiable risk factors for stroke recurrence [5,9], potentially reducing the risk by 25% [10]. Even lower doses of PA than those recommended by the WHO can achieve [11] cardiometabolic benefits [12], reduce sequelae [13], restore independence, and increase participation [5], all contributing to an enhanced quality of life [14]. Therefore, establishing adherence to PA [15] through daily habits and routines is a priority, and current knowledge gaps need to be addressed [16] to include them in the clinical guidelines [17].

On the other hand, stroke survivors face restrictions in participation in everyday activities. The International Classification of Functioning, Disability and Health (ICF) framework [18] states that the term participation involves a variety of life activities, not limited to physical activity or energy-intensive tasks. These restrictions in participation, remain evident even years after the stroke [19]. Specially in social, leisure, work, and home activities, persist despite facilitating factors such as having a wide social network or being functionally independent through ambulation [20]. Multifactorial barriers, including age, family and social support, stroke sequelae, and comorbidities, contribute to these restrictions [21,22].

Although the level of participation contributes to the state of well-being and quality of life [23] and that active participation in everyday activities involves energy expenditure [24], updated information on the relationship between these two therapeutic goals for stroke survivors is still needed. Multifactorial interventions on healthy lifestyle habits have shown greater effects [25] than interventions focusing on single factors. However, a large proportion of studies have focused on the level of PA [6] without considering its relationship with the return to participation, the energy expenditure involved [26], or adherence after rehabilitation translated into habits and behavioural profiles [27]. There is evidence of moderate correlations between PA levels and participation during the six months after a

stroke [28]. However, high-quality methodological studies with higher samples and from different countries are needed to determine if those relationships are retained thereafter the first six months after a stroke. Mild stroke survivors should be prioritised in intervention plans to reduce the socio-economic cost, given their potential to adopt a healthy lifestyle by engaging in physical activity and regaining independence to participate in daily activities [29].

The purpose of this cross-sectional observational study was to analyse the relationships between PA and participation in everyday activities in Spanish mild stroke survivors. Understanding the association between PA and participation may be useful to developing efficient interventions promoting the return to activities with greater benefits.

## Materials and methods

This observational cross-sectional study was part of the Part&Sed-Stroke project, a multi-centre research project focused on lifestyle after a stroke, involving several rehabilitation centres and hospitals across different regions of Spain [29]. Recruitment began on 1st January 2022 and finished on 28th February 2023. The study was approved by the Regional Ethics Committee (PI21/333) and performed in accordance with the Helsinki Declaration. All participants consented to participate in the study before enrolment. This study was conducted according to the STROBE recommendations [30].

### Participants

Stroke survivors who were either currently or had previously received rehabilitation at the collaborating centres and hospitals were enrolled following the recruitment procedure outlined in the Part&Sed-Stroke project protocol [29]. The inclusion criteria were 1) aged more than 18 years; 2) previous stroke event occurring more than 6 months, independent of aetiology; 3) speech and cognitive skills sufficient for communication and understanding (i.e., aphasia absence and a Mini-Mental Cognitive Test score > 24); 4) residing at home; and 5) walking autonomy (i.e., Functional Ambulation Categories ≥ 3). The exclusion criteria were 1) speech or comprehension disturbances for data collection; and 2) inpatient or institutionalised individuals (e.g., nursing homes).

A minimum sample size of at least 100 stroke survivors was initially intended to meet the power requirements. This sample size was determined by an expected correlation coefficient of 0.44, with a confidence interval of ± 0.16 [28] and an alpha value of 0.05 [31]. Missing values about 30% were assumed; therefore, at least 130 participants were requested.

### Outcome measures

**Clinical and sociodemographic data.** Clinical data included the time since stroke, type of stroke, The Functional Ambulation Categories (FAC) for the ambulation ability [32], and the Barthel Index (BI) for functional dependency in activities of daily living [33]. Sociodemographic data consisted of sex, age, and employment status.

**Self-reported physical activity.** The Spanish version of the International Physical Activity Questionnaire – short form (IPAQ-SF) [34] was conducted to evaluated self-reported PA. The IPAQ-SF is a self-report tool for public use (Creative Commons license "CC BY 4.0") that records the duration and frequency of moderate and vigorous PA, walking, and sitting over the past seven days. The output score is expressed in Metabolic Equivalents of Task (METs) minutes a week (MET-min/week) and provides a categorisation of PA levels as low, moderate or high. The IPAQ has demonstrated satisfactory psychometric properties (appropriate content and face validity, construct validity, and excellent test-retest stability for the total

score) in stroke survivors [35]. Furthermore, the use of the questionnaire was supported by a recent expert consensus [36] due to its ability to corroborate compliance with WHO recommendations of weekly PA.

**Objective physical activity.** The activity tracker wristband Fitbit Inspire 2 (*Fitbit, United States, San Francisco, CA*) was used to assess PA levels, considered a valid and reliable method for monitoring PA and sleep hours [37] and recommended by an international consensus for stroke survivors [36]. This device, using a 3-axis accelerometery system, monitors several PA sub-variables. For this study, monitoring was performed for 14 days, and data from 7 full days were chosen according to recommendations for the most sensitive data in relation to moderate to vigorous PA [38]. Average steps/day, and an approximate data of the average calories burned during exercise (daily caloric expenditure) in kilocalories/day (Kcal/day) were collected using heart rate data [36].

**Self-perceived participation level and degree of activity satisfaction.** The participation in everyday activities level, the degree of satisfaction with the activity and participation balance were assessed using the Spanish version of Satisfaction with Daily Occupations-Occupational Balance (SDO-OB) [39] validated for stroke survivors [40]. The SDO-OB demonstrated adequate psychometric properties (acceptable internal consistency, good intra-observer reliability, known group validity, absence of ceiling and floor effect plus error measurement values are available). It covers the nine domains listed in the ICF framework providing support for its use as a reference tool in both clinical and research settings for stroke survivors.

The SDO-OB addressed participation in 13 activities covering four areas: self-care, home management, work and leisure. On each item, the participant stated whether they have participated recently (yes/no) and reported their degree of satisfaction (whether participating or not) with a score from 1 to 7 (1 = being extremely dissatisfied and 7 = being extremely satisfied). The final score is the sum of "Yes" responses to obtain the level of activity participation, ranging from 1 to 7. For the satisfaction score, the points indicated for each item are summed (ranging from 7 to 91) [39]. Data on balance with participation have not been considered in this study.

**Self-perceived retention in participation in everyday activities levels after stroke.** The percentage of retained daily life activities after stroke was scored using the Spanish version of Activity Card Sort (ACS). The ACS covers all participation domains defined by the ICF (i.e., major life areas, community, social and civic life, learning and applying knowledge, general tasks and demands, communication, mobility, self-care, domestic life and interpersonal interactions) [18] except for general task [21]. It records participation in 79 activities divided into four different areas: 26 instrumental activities (e.g., housekeeping, shopping), 23 leisure activities (e.g., listening to music, playing card games), 27 social participation activities (e.g., having a coffee, spending time with friends), and 3 productivity and education activities (e.g., working, studying).

The examiner shows a photograph (i.e., cards) representing each of the 79 activities, and the participant indicates whether the activity was performed before the onset of the stroke. Subsequently, only for the photographs of activities performed, the participant indicates whether they continue to perform the activity as before, perform it less, or no longer perform it. The total score shows the proportion of preserved activities following the stroke compared to their pre-illness activity level. To calculate this score, the four activity areas are combined to indicate the level of engagement with each activity and whether that activity has been discontinued. The ACS is a reliable and validated tool for assessing perceived activity participation levels in Spanish population [41] and has already been successfully used in stroke survivors [42].

## Procedure for the data collection

The data collection involved the principal investigator (CDA, Occupational Therapist and Physiotherapist with over 14 years of experience in neurorehabilitation) and 30 collaborating professionals (19 physiotherapists and 11 occupational therapists with experience in neurore-habilitation) from the 19 centres involved in the study. These collaborators were trained to follow a standardised process for assessment tool implementation and data collection, which were transferred to the principal investigator using the P4Work application in encrypted form for subsequent analysis [43].

The administration protocol for the assessment instruments consisted of grouping the data collection on different occasions over a 14-day period (i.e., at baseline, day 7, and day 14) [29]. Each professional collaborator collected data through face-to-face interviews with participants from their centres, while the principal investigator collected data via videoconference.

On the first day, sociodemographic data and the BI were collected by the professional collaborator, and the Fitbit Inspire 2 wristband was placed on the less affected wrist side for 24-hour monitoring over 14 full days following the recommendations [8,36,44]. After 7 days from the start of data collection, the principal investigator collected clinical data and administered the IPAQ-SF and the SDO-OB via videoconference. At the end of the 14-day follow-up, the collaborating professionals removed the wristbands [29].

## Statistical analysis

All PA and participation variables showed a non-normal distribution after the Kolmogorov-Smirnov test. Quantitative variables were expressed as mean and standard deviations, as well as median and interquartile range due to the non-normal distribution. Categorical variables were expressed as number and percentages. In the case of missing data for a single variable of PA and/or participation, the data for the remaining available variables were used. If both PA and participation variables were missing, the participant was excluded from the analysis.

The correlation between the variables of PA (i.e., MET-min/week, Kcal/day, average steps/day) and participation (i.e., retained activities, participation level and the degree of activity satisfaction) was assessed using Spearman's rank correlation coefficient. Spearman's rho instead of Pearson's r was used due to the non-normal distribution of PA and participation variables [45]. A correlation coefficient was defined as 'negligible' ($rho < 0.1$), 'weak' ($0.10 > rho < 0.39$), 'moderate' ($0.40 > rho < 0.69$), and 'strong' ($rho \geq 0.7$) [28].

Independent forward stepwise multiple regression models were conducted for each PA sub-variable to identify participation sub-variables associated with them. R-squared ($R^2$) was calculated to assess the proportion of variance in the dependent variable explained by the included predictors. Stepwise method instead standard method with all variables was used to avoid overfitting of the model [46].

Statistical analysis was conducted with SPSS v.25 (*IBM, Chicago, IL, USA*). A Bonferroni correction was applied due to multiple comparisons, and significance correlation between compared variables was accepted at $P < 0.01$.

## Results

Table 1 presents the clinical and sociodemographic data of the 123 out of 130 stroke survivors included in the analysis. Table 2 presents the descriptive statistics for the PA and participation variables. Seven participants were excluded from the analysis due to missing data in both PA and participation variables. Of the 123 participants included, complete data were obtained from 109. Missing data from the Fitbit Inspire 2 ($n = 6$) were due to linkage problems

**Table 1. Clinical and sociodemographic data of participants (n = 123).**

|  | Mean ± SD | median [IQR 25–75] |
|---|---|---|
| Age (years) | 61.3 ± 12.4 | 62 [55–70] |
| Female (n, %) | 43, 35% |  |
| Employment status (n, %) |  |  |
| Unpaid work | 3, 2.4% |  |
| Employed work | 3, 2.4% |  |
| Unemployed | 8, 6.5% |  |
| Retired | 47, 38.2% |  |
| Disability pension | 55, 44.7% |  |
| Sick leave | 7, 5.7% |  |
| Time since stroke (months) | 62.4 [62.4] | 41.5 [19–83] |
| Stroke type (n, %) |  |  |
| Haemorrhagic | 52, 42.3% |  |
| Ischemic | 71, 57.7% |  |
| Functional Ambulation Categories |  |  |
| 5[a] | 85,69.1% |  |
| 4[b] | 27, 22.0% |  |
| 3[c] | 11, 8.9% |  |
| Barthel Index (0-100) | 89.6 ± 12.1 | 95 [80–100] |

Data are expressed in Mean ± Standard deviation and median [Interquartile range 25–75] or otherwise specify in number and percentages.

[a]Functional Ambulation Categories-5: Independent ambulator on any surface;

[b]Functional Ambulation Categories-4: Independent ambulator on level surfaces;

[c]Functional Ambulation Categories-3: Dependent ambulator who requires assistance from another person in the form of verbal supervision/guarding.

**Table 2. Descriptive statistics for the physical activity and participation variables.**

|  | Mean ± SD | median [IQR 25–75] |
|---|---|---|
| IPAQ-SF (MET-min/week)[a] | 1275 ± 1150 | 990 [347–1957] |
| Fitbit Inspire 2 (Kcal/day)[b] | 1018 ± 603 | 966 [585–1345] |
| Fitbit Inspire 2 (average steps/day)[b] | 7887 ± 5706 | 6642 [3586–10765] |
| Summed SDO-OB activity level (1–13)[c] | 6.6 ± 2.3 | 7 [5–8] |
| Summed SDO-OB satisfaction (7–91)[c] | 69.5 ± 11.6 | 70 [62–78] |
| ACS total (%)[d] | 62 ± 20 | 64 [48–77] |
| ACS instrumental activities (%)[d] | 62 ± 24 | 66.7 [40–84] |
| ACS leisure activities (%)[d] | 65 ± 22 | 65 [47–82] |
| ACS social participation (%)[d] | 64 ± 22 | 65 [50–81] |
| ACS work (%)[d] | 21 ± 31 | 0 [0–33] |

Data are expressed in Mean ± Standard deviation and median [Interquartile range 25–75].

ACS: Activity Card Sort; SDO-OB: Satisfaction with Daily Occupations-Occupational Balance; IPAQ: International Physical Activity Questionnaire; MET: Metabolic Equivalents of Task.

[a]n = 122;

[b]n = 117;

[c]n = 119;

[d]n = 116.

between the monitoring wristband and the mobile device, as well as non-tolerance of wearing the wristband due to skin reactions or discomfort on the part of the participants. Missing IPAQ-SF (n = 1) and SDO-OB (n = 4) data were due to inability to collect the data, and missing data from ACS were related to problems in dumping and recording the data (n = 7). Full database is available in S1 File.

## Correlations between physical activity and participation assessment tools

Table 3 presents the correlations between the variables of PA (i.e., MET-min/week from the IPAQ-SF; Kcal/day and average steps/day from the Fitbit Inspire 2) and participation (i.e., retained activities from the ACS, and summed participation levels and degree of activity satisfaction from the SDO-OB).

The summed total points of the perception of retained activities showed 'moderate' correlations with Kcal/day and average steps/day (rho ≥ 0.581), and a 'weak' correlation with MET-min/week (rho = 0.324). Regarding the different sub-areas of ACS participation, participation in instrumental activities of daily living had a 'moderate' correlation with the PA sub-variables Kcal/day and average steps/day (rho ≥ 0.640), and a 'weak' correlation with MET-min/week (rho = 0.337). Participation in leisure and social activities had a 'moderate' correlation with both Kcal/day and average steps/day (rho ≥ 0.454), and a 'weak' correlation with MET-min/week (rho = 0.253). In contrast, ACS work activities had a 'weak' correlation with PA levels (rho = 0.248) and a 'negligible' correlation with MET-min/week recorded by IPAQ-SF.

In relation to the summed level of participation, it showed 'moderate' correlations with both Kcal/day and average steps/day (rho ≥ 0.647) and a 'weak' correlation with ME T-min/week (rho = 0.303). The summed activity satisfaction had a 'moderate' correlation with Kcal/day and average steps/day (rho ≥ 0.407) and a 'weak' correlation with MET-min/week (rho = 0.254).

The forward stepwise multiple regression models showed that the ACS instrumental activities and summed SDO-OB activity level explained 39% of the variance ($R^2 = 0.392$) of the average steps/day registered by the activity tracker wristband, as well as 40% ($R^2 = 0.402$) of the Kcal/day. In contrast, the summed SDO-OB activity level explained only 6% of the variance ($R^2 = 0.063$) of the MET-min/week in the IPAQ-SF.

**Table 3. Table of correlations between physical activity and participation variables.**

|  | IPAQ-SF (MET-min/week) | Fitbit Inspire 2 (Kcal/day) | Fitbit Inspire 2 (average steps/day) |
|---|---|---|---|
| Summed SDO-OB activity level | 0.30** | 0.66** | 0.65** |
| Summed SDO-OB activity satisfaction | 0.25* | 0.43** | 0.41** |
| ACS total | 0.32** | 0.59** | 0.58** |
| ACS instrumental activities | 0.34** | 0.65** | 0.64** |
| ACS leisure activities | 0.25* | 0.45** | 0.46** |
| ACS social participation | 0.30* | 0.47** | 0.47** |
| ACS work | 0.14 | 0.28* | 0.25* |

Data are expressed in Spearman correlation coefficient.

ACS: Activity Card Sort; SDO-OB: Satisfaction with Daily Occupations-Occupational Balance; IPAQ: International Physical Activity Questionnaire; MET: Metabolic Equivalents of Task.

*p < 0.01;

**p < 0.001.

### Correlations between physical activity assessment tools

This study found a 'strong' correlation (rho = 0.933) between the sub-variables of PA average steps/day and Kcal/day. For MET-min/week a 'moderate' correlation was found with average steps/day (rho = 0.479) and Kcal/day (rho = 0.406).

### Correlations between participation assessment tools

This study found a 'moderate' correlation between the summed total score of retained activities and the summed levels of participation (rho = 0.542) and the summed activity satisfaction (rho = 0.448).

## Discussion

This observational cross-sectional study in mild stroke survivors at least 6 months post-stroke found bilateral relationship with their current participation levels and the retained instrumental activities of daily living. While the objective PA sub-variables showed stronger correlations than self-perceived PA variables, weaker correlations were found with leisure and social participation sub-variables compared to participation in instrumental activities.

### Correlation between physical activity and participation sub-variables

The findings of this study have confirmed the hypothesis that mild stroke survivors with higher levels of PA also present high levels of participation [47,48]. This correlation has several implications that are discussed below.

 The most relevant associations in this study were found between the variables of PA and participation in instrumental activities (e.g., shopping or cleaning the house). This reinforces that active participation in everyday activities involves energy expenditure [24]. In this sense, the active participation could potentially contribute to the benefits typically ascribed to PA and related to the mitigation of risk factors for recurrent stroke such as hyperlipidaemia, hypertension, or obesity [49–51], especially in mild stroke survivors. In contrast, it is logical that participation in leisure activities (e.g., going to the cinema or having a coffee) or the level of activity satisfaction showed a smaller association with the level of PA, considering the tendency of stroke survivors to reduce physical demanding leisure activities and to spend more sedentary leisure time [20,52,53]. Given these results, a strategy to increase PA levels in stroke survivors performing only leisure activities and reluctant to exercise could be to increase the variety of the participation activities, selecting those that involve more energy expenditure. Nevertheless, further studies are needed to determine the multifactorial relationships with other variables and their importance in the lifestyle of stroke survivors [19,53–56].

 The 'moderate' correlation found between retained leisure and social activities and average steps/day and Kcal/day highlights the relationship between walk fitness and participation in the community of mild stroke survivors [57–59]. Moreover, the 'moderate' correlation between SDO-OB and the objective PA variables corresponds with the results found by Sullivan et al. [60] in stroke survivors over 6 months post-stroke. This relationship suggests that information on PA sub-variable can predict for future levels of participation of stroke survivors.

 Additionally, the correlations between levels of PA and participation found in this study in stroke survivors more than 6 months post-stroke support the findings of a recent systematic review [28], with the novelty of incorporating appropriate assessment tools that cover several sub-variables of the PA and participation domains.

## Correlation within assessment tools

Regarding the assessment tools for PA levels, the 'strong' correlation found between Kcal/day and average steps/day, as well as the 'moderate' correlation between these values and MET-min/week, support the use of these PA sub-variables in stroke survivors for an easy PA assessment. Specifically, these PA sub-variables can determine both the level of PA and adherence to WHO recommendations [61]. Furthermore, recent clinical guidelines have proposed that the compliance with WHO recommendations can be measured not only by recording minutes and intensity of PA, but also by PA sub-variables like Kcal/week [62,63], or average of steps/day [64]. Moreover, correlations between participation level and PA levels were more accurately estimated with the Fitbit Inspire 2 compared to the IPAQ-SF, supporting monitoring devices as more reliable methods than self-reported questionnaires [65]. However, self-reported questionnaires are recommended and useful in the absence of these devices [36].

Regarding the assessment tools for participation, both the summed level of participation recorded by the SDO-OB and the perception of retained activities provided by the ACS showed a 'moderate' correlation. This relationship between both tools, which have been previously recommended to provide information on the complexity of participation [66,67], implies that the use of either of these two scales would independently provide valuable information on participation levels post-stroke.

## Clinical implications and future perspectives

Intervention plans for stroke survivors should consider the contribution of retained activities of daily living (especially instrumental ones) and its correlation with PA levels to a healthy lifestyle. Moreover, the daily environments in which these activities are performed, as well as individual preferences that reinforce adherence to these activities [68], must also be considered to establish a healthy daily routine [69] and mitigate healthcare costs associated with patient dependency [70]. However, the moderate correlations between most of the PA and participation variables suggest that at the clinical level it may be insufficient to encourage only an increase in PA without its integration into participation in everyday activities of the stroke survivor. Therefore, physical therapy programs should take advantage of the participation activities that their patients enjoy in order to promote healthy levels of PA and adherence to it.

Both self-reported questionnaires and objective monitoring devices are reliable and useful assessment methods, although the latter is a more accurate one. If an activity monitoring device, such as the Fitbit Inspire 2, is not available in a clinical setting, the data calculated from MET-minutes per week using the IPAQ-SF can be considered a suitable screening reference tool. This approach is adequate for identifying individuals not complying with the WHO recommendations on PA [36]. However, PA variables from objective devices (e.g., Fitbit Inspire 2), compared to self-reported PA questionnaires (e.g., IPAQ-SF), are explained up to 40% by participation sub-variables.

Future studies should explore the multifactorial relationships involved in the lifestyle of stroke survivors and identify predictors of higher levels of PA and participation in these individuals.

## Limitations

Data on 'moderate and vigorous' PA minutes provided by both the IPAQ-SF and the Fitbit Inspire 2 were not used because, for most of participants, they provided "zero values" which impeded statistical analysis. Therefore, only the PA sub-variables Kcal/day, average steps/day, and MET-min/week were used. Additionally, despite the participants in this study belonged to a group of stroke survivors with a good functional level and an average age of 61 years, a large

percentage did not return to work. Thus, it was not possible to explore relationships between participation in these activities and levels of PA, highlighting the need for further studies in this respect.

## Conclusions

The level of PA self-reported by the IPAQ-SF and monitored with the Fitbit Inspire 2 device in mild stroke survivors shows a bilateral correlation with their retained instrumental activities of daily living participation measured with the ACS and their participation in everyday activities level measured with the SDO-OB. Overall, restoring participation in everyday activities levels after stroke does not depend exclusively on PA levels, and other factors need to be considered in future studies. Furthermore, it may be suggested that rehabilitation programs must not focus only on PA without considering the context within daily life activities in which PA can be integrated.

## Supporting information

**S1 File. Database.**
(SAV)

## Acknowledgments

We thank the following professional collaborators for their participation in the development of the pilot research: Inés Cortés Cabeza; Ana Conte Lamenca; Leyre Leceaga Gaztambide; Carina Francisco Salgueiro; Lucía Díez Fuentes; Laura Ares Barge; and Beatriz Martín Lamata.

We also thank all the Spanish collaborating centers and professionals involved in the recruitment of study participants: ADACECO, AENO, AGREDACE, AIDA, ASPAYM, CENNER, CIRON, Clínica de neurorehabilitación, Fundación Pita López, Grupo 5 CIAN Navarra, Grupo 5 CIAN Zaragoza,Hospital Provincial Sagrado Corazón de Jesús, Hospital Universitario San Jorge, INEURO, NEUFIS, NEURAXIS, NEUROESPLUGUES, and Centro de fisioterapia El Carmen, as well as the self-employed professionals Lezcano Fisioterapia and Juan Luis Abeledo Alcón.

'Part&Sed-Stroke collaborators' list of names:

Alicia, Tornero Navarro; Ana, Conte Lamenca; Andrea, Yerro Astrain; Beatriz, Martín Lamata; Carina, Salgueiro; Claudia, Marín Marín; Diana, Ruiz Ramos; Elena, Sanz Sanza; Enrique, Villa Berges; Fernando, Cuesta Ruiz; Inés, Cortés Cabeza; Javier, Harguindey; Juan Luis, Abeledo Alcón; Laura, Ares Barge; Leyre, Leceaga Gaztambide; Lilian, Le Roux; Lourdes, Martín Gros; Lucía, Díez Fuentes; María Carmen, Grácia Sen; María Pilar, Pardo Sanz; Natalia Muiño, Paloma, Rodríguez Escudero; Paz Cristina, Sánchez Lecina; Rebeca, Yebra Vilarchao; Sara, Beltrán Roche; Verónica, Montoya Murillo.

## Author contributions

**Conceptualization:** Cristina de Diego-Alonso, Jorge Alegre-Ayala, Julia Blasco-Abadía, Víctor Doménech-García, Pablo Bellosta-López.

**Data curation:** Pablo Bellosta-López.

**Formal analysis:** Cristina de Diego-Alonso, Pablo Bellosta-López.

**Funding acquisition:** Cristina de Diego-Alonso, Pablo Bellosta-López.

**Investigation:** Cristina de Diego-Alonso.

**Methodology:** Cristina de Diego-Alonso, Pablo Bellosta-López.

**Project administration:** Cristina de Diego-Alonso.

**Resources:** Cristina de Diego-Alonso, Jorge Alegre-Ayala, Víctor Doménech-García, Pablo Bellosta-López.

**Software:** Jorge Alegre-Ayala.

**Supervision:** Pablo Bellosta-López.

**Validation:** Julia Blasco-Abadía, Víctor Doménech-García.

**Visualization:** Cristina de Diego-Alonso, Julia Blasco-Abadía, Pablo Bellosta-López.

**Writing – original draft:** Cristina de Diego-Alonso.

**Writing – review & editing:** Jorge Alegre-Ayala, Julia Blasco-Abadía, Víctor Doménech-García, Pablo Bellosta-López.

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
