## [Decision Letter · Decision Letter 0]

26 Dec 2024

PONE-D-24-44214Association between objective and self-perceived levels of physical activity and participation in daily life activities in mild stroke survivors.PLOS ONE

Dear Dr. de Diego-Alonso,

Thank you for submitting your manuscript to PLOS ONE. After careful consideration, we feel that it has merit but does not fully meet PLOS ONE’s publication criteria as it currently stands. Therefore, we invite you to submit a revised version of the manuscript that addresses the points raised during the review process.

We look forward to receiving your revised manuscript.

Kind regards,

Sohel Ahmed, MPT, MDMR

Academic Editor

PLOS ONE

Journal Requirements:

-Prevalence and correlates of stroke among older adults in Ghana: Evidence from the Study on Global AGEing and adult health

(SAGE)  (https://doi.org/10.1371/journal.pone.0212623)

(among others)

In your revision ensure you cite all your sources (including your own works), and quote or rephrase any duplicated text outside the methods section. Further consideration is dependent on these concerns being addressed.

3. In the online submission form you indicate that your data is not available for proprietary reasons and have provided a contact point for accessing this data. Please note that your current contact point is a co-author on this manuscript. According to our Data Policy, the contact point must not be an author on the manuscript and must be an institutional contact, ideally not an individual. Please revise your data statement to a non-author institutional point of contact, such as a data access or ethics committee, and send this to us via return email. Please also include contact information for the third party organization, and please include the full citation of where the data can be found.

Reviewers' comments:

Reviewer's Responses to Questions

**Comments to the Author**

1. Is the manuscript technically sound, and do the data support the conclusions?

Reviewer #1: Yes

Reviewer #2: Yes

2. Has the statistical analysis been performed appropriately and rigorously? 

Reviewer #1: Yes

Reviewer #2: No

3. Have the authors made all data underlying the findings in their manuscript fully available?

Reviewer #1: Yes

Reviewer #2: Yes

4. Is the manuscript presented in an intelligible fashion and written in standard English?

Reviewer #1: Yes

Reviewer #2: Yes

5. Review Comments to the Author

Reviewer #1: Good paper with enough information. However, there are some points that need to be modified. Firstly, please omit the "smple size.". In method section. It was repeated. Secondly, please explain more about outcome measures and the tools that assess them.

Reviewer #2: Thank you for the opportunity to review the manuscript, ‘Association between objective and self-perceived levels of physical activity and participation in daily life activities in mild stroke survivors. Though it is an interesting article, there are numerous methodological issues. The following references can improve the quality of the manuscript.

1. Line 49 states mild stroke survivors. Why were only mild stroke survivors included in the study?

2. Was the choice of the International Physical Activity Questionnaire (IPAQ-SF) based on its relevance?

3. Line No. 137-138 Statement: "The IPAQ has demonstrated satisfactory psychometric properties in stroke survivors." The authors are encouraged to explain in the manuscript.

4. Line 50 in the abstract and Lines 275–278 in the discussion state different things. Please clarify? 6. Could authors clarify on which basis quantitative variables were expressed as mean and standard deviations and/or median depending on the data distribution? As not mention about any test of normality used or not.

7. In the statistical analysis section, lines 215–217, please provide an explanation for using the Spearman rank correlation (non-parametric test) and why and why not the parametric test (Pearson correlation coefficient test).

8. Could authors clarify why not using multiple regression/logistic regression to find out association or relationship between PA and participation in daily life activities in Spanish mild stroke survivors?

6. PLOS authors have the option to publish the peer review history of their article (what does this mean? ). If published, this will include your full peer review and any attached files.

**Do you want your identity to be public for this peer review?** For information about this choice, including consent withdrawal, please see our Privacy Policy .

Reviewer #1: **Yes: ** Laleh Abadi marand

Reviewer #2: No

---

## [Author Response · Author response to Decision Letter 1]

7 Feb 2025

Reviewer(s)' Comments to Author:

Reviewer #1:

Good paper with enough information. However, there are some points that need to be modified.

We are very grateful for the reviewer's comments on the manuscript.

Firstly, please omit the "smple size.". In method section. It was repeated.

We appreciate the reviewer's comment. The erratum has been corrected (on page 4, line 141-145):

“A minimum sample size of at least 100 stroke survivors was initially intended to meet the power requirements. This sample size was determined based on an expected correlation coefficient of 0.44, with a confidence interval of ±0.16[28] and an alpha value of 0.05[31]. Missing values about 30% were assumed.; therefore, at least 130 participants were requested.”

Secondly, please explain more about outcome measures and the tools that assess them.

We appreciate the reviewer's comments on this section and agree with the points raised.

The information has been included in the manuscript (on page 4, 5 and 6):

”Clinical data included the time since stroke, type of stroke, The Functional Ambulation Categories (FAC) for the ambulation ability[32], and the Barthel Index (BI) for functional dependency in activities of daily living[33]. Sociodemographic data consisted of sex, age, and employment status.”

“The Spanish version of the International Physical Activity Questionnaire – short form (IPAQ-SF)[34] was conducted to evaluated self-reported PA. The IPAQ-SF is a self-report tool for public use (Creative Commons license “CC BY 4.0”) that records the duration and frequency of moderate and vigorous PA, walking, and sitting over the past seven days. The output score is expressed in Metabolic Equivalents of Task (METs) minutes a week (MET-min/week) and provides a categorisation of PA levels as low, moderate or high. The IPAQ has demonstrated satisfactory psychometric properties (appropriate content and face validity, construct validity, and excellent test-retest stability for the total score) in stroke survivors[35] and it was supported by a recent expert consensus[36] due to its ability to corroborate compliance with WHO recommendations of weekly PA.”

“The activity tracker wristband Fitbit Inspire 2 (Fitbit, United States, San Francisco, CA) was used to assess PA levels, considered a valid and reliable method for monitoring PA and sleep hours[37] and recommended by an international consensus for stroke survivors[36]. This device, using a 3-axis accelerometery system, monitors several PA sub-variables. For this study, monitoring was performed for 14 days, and data from 7 full days were chosen according to recommendations for the most sensitive data in relation to moderate to vigorous PA[38]. Average steps/day, and an approximate data of the average calories burned during exercise (daily caloric expenditure) in kilocalories/day (Kcal/day) were collected using heart rate data[36].”

“The participation in everyday activities level, the degree of satisfaction with the activity and participation balance were assessed using the Spanish version of Satisfaction with Daily Occupations-Occupational Balance (SDO-OB)[39] validated for stroke survivors[40]. The SDO-OB demonstrated adequate psychometric properties (acceptable internal consistency, good intra-observer reliability, known group validity, absence of ceiling and floor effect plus error measurement values are available). It covers the nine domains listed in the ICF framework providing support for its use as a reference tool in both clinical and research settings for stroke survivors.

The SDO-OB addressed participation in 13 activities covering four areas: self-care, home management, work and leisure. On each item, the participant stated whether they have participated recently (yes/no) and reported their degree of satisfaction (whether participating or not) with a score from 1 to 7 (1=being extremely dissatisfied and 7=being extremely satisfied). The final score is the sum of “Yes” responses to obtain the level of activity participation, ranging from 1 to 7. For the satisfaction score, the points indicated for each item are summed (ranging from 7 to 91)[39]. Data on balance with participation have not been considered in this study.”

“The ACS is a reliable and validated tool for assessing perceived activity participation levels in Spanish population[41] and has already been successfully used in stroke survivors[42]”

An erratum has been detected and corrected (on page 6, line 232):

“On the first day, sociodemographic data and the BI were collected by the professional collaborator, and the Fitbit Inspire 2 wristband was placed on the less affected wrist side for 24-hour monitoring over 14 full days following the recommendations”

Reviewer #2:

Thank you for the opportunity to review the manuscript, ‘Association between objective and self-perceived levels of physical activity and participation in daily life activities in mild stroke survivors. Though it is an interesting article, there are numerous methodological issues. The following references can improve the quality of the manuscript.

1. Line 49 states mild stroke survivors. Why were only mild stroke survivors included in the study?

We thank the reviewer for pointing this question out and agree to provide information to clarify it.

The sample of this study belongs to the multicentre research project Part&Sed-Stroke. This project focused on analysing a section of the stroke survivor population that, due to the mild sequelae, should not present difficulties in leading a healthy lifestyle in relation to physical activity and be able to resume participation.

These mild sequeale participants could potentially be a segment of stroke survivors who would not incur a socio-economic cost if they follow a healthy lifestyle based on reaching physical activity recommendations and returning to independent participation in daily activities.

Furthermore, in several of the most representative studies that have previously analyzed participation restrictions of people with chronic stroke, the majority of participants had mild severity. We show the percentages or means/medians according to the NIHSS scale of the participants in these studies:

• Eriksson et al., (2012), percentage of mild stroke (76%).

• De Graaf et al., (2018), percentage of minor stroke (57%).

• De Graaf et al., (2023), mean of stroke severity based on NIHSS (4= minor stroke).

• De Graaf et al., (2022), percentage of minor stroke (56%).

• Palstam et al., (2019), percentage very mild stroke (67%).

• Svensson et al., (2019), percentage very mild stroke (63%); percentage mild stroke (10%).

• Norlander et al., (2016), median of stroke severity based on NIHSS (3=minor stroke).

• Verberne et al. (2019), percentage of mild stroke (56%).

• Singam et al., (2015), percentage of mild stroke (79%).

References:

Eriksson G, Aasnes M, Tistad M, Guidetti S, von Koch L. Occupational gaps in everyday life one year after stroke and the association with life satisfaction and impact of stroke. Top Stroke Rehabil. 2012 May-Jun;19(3):244-55. doi: 10.1310/tsr1903-244. PMID: 22668679.

de Graaf JA, van Mierlo ML, Post MWM, Achterberg WP, Kappelle LJ, Visser-Meily JMA. Long-term restrictions in participation in stroke survivors under and over 70 years of age. Disabil Rehabil. 2018 Mar;40(6):637-645. doi: 10.1080/09638288.2016.1271466. Epub 2017 Jan 5. PMID: 28054834.

de Graaf JA, Wondergem R, Kooijmans ECM, Pisters MF, Schepers VPM, Veenhof C, Visser-Meily JMA, Post MWM. The longitudinal association between movement behavior patterns and the course of participation up to one year after stroke. Disabil Rehabil. 2023 Aug;45(17):2787-2795. doi: 10.1080/09638288.2022.2109071. Epub 2022 Aug 9. PMID: 35944521.

de Graaf JA, Schepers VPM, Nijsse B, van Heugten CM, Post MWM, Visser-Meily JMA. The influence of psychological factors and mood on the course of participation up to four years after stroke. Disabil Rehabil. 2022 May;44(10):1855-1862. doi: 10.1080/09638288.2020.1808089. Epub 2020 Aug 31. PMID: 32866072.

Palstam A, Sjödin A, Sunnerhagen KS. Participation and autonomy five years after stroke: A longitudinal observational study. PLoS One. 2019 Jul 8;14(7):e0219513. doi: 10.1371/journal.pone.0219513. PMID: 31283800; PMCID: PMC6613678.

Svensson JS, Westerlind E, Persson HC, Sunnerhagen KS. Occupational gaps 5 years after stroke. Brain Behav. 2019 Mar;9(3):e01234. doi: 10.1002/brb3.1234. Epub 2019 Feb 19. PMID: 30784220; PMCID: PMC6422817.

Norlander A, Carlstedt E, Jönsson AC, Lexell EM, Ståhl A, Lindgren A, Iwarsson S. Long-Term Predictors of Social and Leisure Activity 10 Years after Stroke. PLoS One. 2016 Feb 22;11(2):e0149395. doi: 10.1371/journal.pone.0149395. PMID: 26901501; PMCID: PMC4765767.

Verberne DPJ, Post MWM, Köhler S, Carey LM, Visser-Meily JMA, van Heugten CM. Course of Social Participation in the First 2 Years After Stroke and Its Associations With Demographic and Stroke-Related Factors. Neurorehabil Neural Repair. 2018 Sep;32(9):821-833. doi: 10.1177/1545968318796341. Epub 2018 Sep 4. PMID: 30178696; PMCID: PMC6146317.

Singam A, Ytterberg C, Tham K, von Koch L. Participation in Complex and Social Everyday Activities Six Years after Stroke: Predictors for Return to Pre-Stroke Level. PLoS One. 2015 Dec 10;10(12):e0144344. doi: 10.1371/journal.pone.0144344. PMID: 26658735; PMCID: PMC4692261.

This methodological design should be replicated in other segments of stroke survivors at different stages of their process and with different degrees of sequelae in order to analyse the situation in the remaining population.

In the ‘introduccion’ section, the following information has been included (on page 3, line 114-116)

“Mild stroke survivors should be prioritised in intervention plans to reduce the socio-economic cost, given their potential to adopt a healthy lifestyle by engaging in physical activity and regaining independence to participate in daily activities[29]”

2. Was the choice of the International Physical Activity Questionnaire (IPAQ-SF) based on its relevance?

IPAQ is a questionnaire that has recently been recommended by an expert consensus (Fini NA 2023). The IPAQ-SF is low-cost and requires less time to administer as it can report useful information through fewer items, reducing sample overload by collecting information on several variables.

Additionally, the IPAQ-SF has recently been shown to be a valid tool to determine the level of self-perceived physical activity and provide information on adherence to WHO physical activity recommendations in stroke survivors, as well as its correlation with objective devices (de Diego-Alonso C 2024).

References:

Fini NA, Simpson D, Moore SA, Mahendran N, Eng JJ, Borschmann K, Moulaee Conradsson D, Chastin S, Churilov L, English C. How should we measure physical activity after stroke? An international consensus. Int J Stroke. 2023 Oct;18(9):1132-1142. doi: 10.1177/17474930231184108. Epub 2023 Jun 24. PMID: 37300499; PMCID: PMC10614172.

de Diego-Alonso C, Blasco-Abadía J, Doménech-García V, Bellosta-López P. Validity and stability of the international physical activity questionnaire short-form for stroke survivors with preserved walking ability. Top Stroke Rehabil. 2024 Oct 22:1-10. doi: 10.1080/10749357.2024.2417645. Epub ahead of print. PMID: 39436814.

3. Line No. 137-138 Statement: "The IPAQ has demonstrated satisfactory psychometric properties in stroke survivors." The authors are encouraged to explain in the manuscript.

We are grateful for the reviewer's proposal. The information has been included in the manuscript together with the appropriate reference (on page 5, line 159-165).

“The output score is expressed in Metabolic Equivalents of Task (METs) minutes a week (MET-min/week) and provides a categorisation of PA levels as low, moderate or high. The IPAQ has demonstrated satisfactory psychometric properties (appropriate content and face validity, construct validity, and excellent test-retest stability for the total score) in stroke survivors[35]. Furthermore, the use of the questionnaire was supported by a recent expert consensus[36] due to its ability to corroborate compliance with WHO recommendations of weekly PA.”

4. Line 50 in the abstract and Lines 275–278 in the discussion state different things. Please clarify?

We thank the reviewer for pointing out this discrepancy. We have rewritten the first paragraph of the discussion to improve clarity and consistency. It now reads as follows (on page 10, line 317-322):

“This observational cross-sectional study in mild stroke survivors at least 6 months post-stroke found bilateral relationship with their current participation levels and the retained instrumental activities of daily living. While the objective PA sub-variables showed stronger correlations than self-perceived PA variables, weaker correlations were found with leisure and social participation sub-variables.”

6. Could authors clarify on which basis quantitative variables were expressed as mean and standard deviations and/or median depending on the data distribution? As not mention about any test of normality used or not.

We thank the reviewer for pointing this out. In our previous version we missed writing that we had performed the Kolmogorov-Smirnov test to explore the normal distribution of the data.

We have modified the text as follows (on page 6, line 240-242):

“All PA and participation variables showed a non-normal distribution after the Kolmogorov-Smirnov test. Quantitative variables were expressed as mean and standard deviations, as well as median and interquartile range due to the non-normal distribution.”

7. In the statistical analysis section, lines 215–217, please provide an explanation for using the Spearman rank correlation (non-parametric test) and why and why not the parametric test (Pearson correlation coefficient test).

In line with our previous response, we have added a sentence in the statistics section to clarify that Spearman rank correlation test was chosen due to the non-parametric distribution of the PA and participation data (on page 7, line 247-251):

“The correlation between the variables of PA (i.e., MET-min/week, Kcal/day, average steps/day) and participation (i.e., retained activities, participation level and the degree of activity satisfaction) was assessed using Spearman´s rank correlation coefficient. Spearman’s rho instead of Pearson’s r was used due to the non-normal distribution of PA and participation variables[45]”.

8. Could authors clarify why not using multiple regression/logistic regression to find out association or relationship between PA and participation in daily life activities in Spanish mild stroke survivors?

We thank reviewer for such pertinent comment. This study focused on providing absent information on stroke survivors living in Spain with mild sequelae regarding the correlation between the variables physical activity and participation following the need identified in the systematic review research study (de Diego-Alonso C 2024). Nevertheless, we agree that presenting a multiple regression analysis could provide further relevant information about the relationship between PA and participation variables. We have performed the following changes in the actions of Statistics (on page 7, line 254-258), Results (on page 10, line 300-304) and Clinical implications (on page 12, line 393-395).

“Independent forward stepwise multiple regression models were conducted for each PA sub-variable to identify participation sub-variables associated with them. R-squared (R²) was calculated to assess the proportion of variance in the dependent variable explained by the included predictors. Stepwise method instead standard method with all variables was used to avoid overfitting of the model[46].”

“The forward stepwise multiple regression models showed that the ACS instrumental activities and summed SDO-OB activity level explained 39% of the variance (R²=0.392) of the average steps/day registered by the activity tracker wristband, as well as 40% (R²=0.402) of the Kcal/day. In contrast, the summed SDO-OB activity level explained only 6% of the variance (R²=0.063) of the MET-min/week in the IPAQ-SF.”

“However, PA variables from objective devices (e.g., Fitbit

---

## [Decision Letter · Decision Letter 1]

28 Feb 2025

Associations between objective and self-perceived physical activity and participation in everyday activities in mild stroke survivors

PONE-D-24-44214R1

Dear Dr. Diego Alonso

We’re pleased to inform you that your manuscript has been judged scientifically suitable for publication and will be formally accepted for publication once it meets all outstanding technical requirements.

Within one week, you’ll receive an email detailing the required amendments. When these have been addressed, you’ll receive a formal acceptance letter, and your manuscript will be scheduled for publication.

An invoice will be generated when your article is formally accepted. Please note that if your institution has a publishing partnership with PLOS and your article meets the relevant criteria, all or part of your publication costs will be covered. Please make sure your user information is up-to-date by logging into Editorial Manager at Editorial Manager®  and clicking the ‘Update My Information' link at the top of the page. If you have any questions relating to publication charges, please contact our author billing department directly at authorbilling@plos.org.

If your institution or institutions have a press office, please notify them about your upcoming paper to help maximize its impact. If they’ll be preparing press materials, please inform our press team as soon as possible—no later than 48 hours after receiving the formal acceptance. Your manuscript will remain under strict press embargo until 2 pm Eastern Time on the date of publication. For more information, please contact onepress@plos.org.

Kind regards,

Sohel Ahmed, BPT, MPT, MDMR

Academic Editor

PLOS ONE

Additional Editor Comments (optional):

Reviewers' comments:

Reviewer's Responses to Questions

**Comments to the Author**

1. If the authors have adequately addressed your comments raised in a previous round of review and you feel that this manuscript is now acceptable for publication, you may indicate that here to bypass the “Comments to the Author” section, enter your conflict of interest statement in the “Confidential to Editor” section, and submit your "Accept" recommendation.

Reviewer #1: All comments have been addressed

Reviewer #2: All comments have been addressed

2. Is the manuscript technically sound, and do the data support the conclusions?

Reviewer #1: Yes

Reviewer #2: Yes

3. Has the statistical analysis been performed appropriately and rigorously? 

Reviewer #1: Yes

Reviewer #2: Yes

4. Have the authors made all data underlying the findings in their manuscript fully available?

Reviewer #1: Yes

Reviewer #2: Yes

5. Is the manuscript presented in an intelligible fashion and written in standard English?

Reviewer #1: Yes

Reviewer #2: Yes

6. Review Comments to the Author

Reviewer #1: The article is complete and acceptable, and in my opinion. All of the comments have been addressed in the revised paper.

Reviewer #2: (No Response)

7. PLOS authors have the option to publish the peer review history of their article (what does this mean? ). If published, this will include your full peer review and any attached files.

**Do you want your identity to be public for this peer review?** For information about this choice, including consent withdrawal, please see our Privacy Policy .

Reviewer #1: **Yes: ** Laleh Abadi marand

Reviewer #2: No

---

## [Editor Report · Acceptance letter]

PONE-D-24-44214R1

PLOS ONE

Dear Dr. de Diego-Alonso,

I'm pleased to inform you that your manuscript has been deemed suitable for publication in PLOS ONE. Congratulations! Your manuscript is now being handed over to our production team.

Kind regards,

on behalf of

Dr. Sohel Ahmed

Academic Editor

PLOS ONE